# A Diet Rich in Essential Amino Acids Inhibits the Growth of HCT116 Human Colon Cancer Cell In Vitro and In Vivo

**DOI:** 10.3390/ijms26147014

**Published:** 2025-07-21

**Authors:** Giovanni Corsetti, Claudia Romano, Silvia Codenotti, Evasio Pasini, Alessandro Fanzani, Tiziano Scarabelli, Francesco S. Dioguardi

**Affiliations:** 1Department of Clinical and Experimental Sciences, University of Brescia, 25121 Brescia, Italy; cla300482@gmail.com (C.R.); evpasini@gmail.com (E.P.); 2Department of Molecular and Translational Medicine, University of Brescia, 25121 Brescia, Italy; silvia.codenotti@unibs.it (S.C.); alessandro.fanzani@unibs.it (A.F.); 3Institute of Human Health, Ruaha Catholic University, Iringa P.O. Box 774, Tanzania; 4Holy Cross Medical Center, Taos, NM 87571, USA; tscarabelli@hotmail.com; 5Nutri-Research s.r.l., 20127 Milano, Italy; fsdioguardi@gmail.com

**Keywords:** cancer, colon, human, HCT116, essential amino acids, BCAA, food, mice

## Abstract

The metabolic hyperactivity of tumor cells demands a substantial amount of energy and molecules to build new cells and expand the tumor, diverting these resources from healthy cells. Amino acids (AAs) are the only totipotent and essential molecules for protein construction. Previous in vitro studies in human and murine cancer cells, along with in vivo studies in mice, have shown that an excess of essential amino acids (EAAs) exerts an inhibitory effect on tumor proliferation by promoting apoptosis and autophagy. In this study, both in vitro and in vivo, we evaluated whether a mixture based on EAA can influence the development of human colon cancer (HCT116). To this end, in vitro, we assessed the proliferation of HCT116 cells treated with a special mix of EAA. In vivo, immunosuppressed athymic nude mice, injected with HCT116 cells subcutaneously (s.c.) or intraperitoneally (i.p.), were given a modified EAAs-rich diet (EAARD) compared to the standard laboratory diet (StD). In vitro data showed that the EAA mix impairs cancer growth by inducing apoptosis and autophagy. In vivo, the results demonstrated that EAARD-fed mice developed s.c. tumors significantly smaller than those of StD-fed mice (total mass 3.24 vs. 6.09 g, respectively). Mice injected i.p. and fed with EAARD showed a smaller and more limited number of intra-peritoneal tumors than StD-fed mice (total mass 0.79 vs. 4.77 g, respectively). EAAs prevents the growth of HCT116 cells by inducing autophagy and apoptosis, increasing endoplasmic reticulum stress, and inhibiting inflammation and neo-vascularization. In addition, the EAARD-fed mice, maintained muscle mass and white and brown adipose tissues. A diet with an excess of EAAs affects the survival and proliferative capacity of human colon cancer cells, maintaining anabolic stimuli in muscular cells.

## 1. Introduction

According to the World Health Organization, colorectal cancer is a prevalent digestive tract malignancy and one of the leading cancers worldwide, yet it is also considered one of the most preventable cancers [1]. The incidence and mortality rates of colorectal cancer increase with the progressive adoption of Western lifestyles [2]. Cancer development is highly influenced by dietary habits, particularly the excessive consumption of refined sugars, animal proteins, fats, and salt, which are characteristic of Western diets. This highlights cancer as a disease dependent on the quality and availability of metabolic substrates [3].

Cancer cells, due to their rapid growth, must adapt their metabolism to increase biomass and ATP production while maintaining redox balance. This metabolic reprogramming is crucial to sustain hyper-proliferation, especially in nutrient-limited environments. Altering or disrupting these processes, such as through the dysregulation of glucose metabolism, could potentially slow tumor growth. However, an elevated demand for amino acids (AAs) is essential to support cancer cell proliferation in various tumors [4,5]. As a result, tumors are particularly susceptible to variations in AAs concentrations and their homeostatic mechanisms [6]. Therefore, gaining a better understanding of AAs metabolism in cancer cells, as well as how these cells utilize and sense nutrients, represents a promising avenue for studying metabolic changes in cancer and improving potential therapeutic strategies.

Previous studies investigating the anti-tumor properties of specific nutrients have examined the effects of a diet containing lysine, proline, arginine, ascorbic acid, and green tea extract on the growth of human colon HCT116 cancer cells in athymic nude mice, suggesting the potential of nutrient combinations as anticancer agents [7]. Based on this principle, numerous approaches have been explored to inhibit or kill tumor cells by obstructing various metabolic pathways. Among these, the metabolic networks of all amino acids (AAs) are complex and interlinked with other pathways [8,9,10]. Under genotoxic, oxidative, and nutritional stress, AAs can facilitate cancer cell survival and proliferation, making AAs metabolism a promising therapeutic target for cancer patients [11].

AAs are generally categorized into essential and non-essential types. Essential amino acids (EAAs) must be obtained through diet, as mammals have limited capacity to synthesize them autonomously, leading to serious metabolic consequences if deficient. In contrast, non-essential amino acids (NEAAs) are abundant in foods and can be synthesized by the body from EAAs when needed. The adequate availability of all EAAs is critical for protein synthesis [12].

In tumor cells, AAs metabolism is altered to support the anabolic processes essential for rapid proliferation [8,11]. These cells “cannibalize” the body’s tissues, particularly collagen and skeletal muscle proteins, to obtain the necessary AAs, resulting in a hyper-catabolic state and cachexia. Cancer cachexia significantly worsens patient resistance to chemotherapy, increases its toxicity, and reduces quality of life and survival [13]. Thus, adequate nutrition and EAAs supplementation in cancer patients can counteract cachexia and mitigate chemotherapy’s collateral damage [14].

Supplementation with an EAA-rich mixture has been effective in hindering tumor progression by inhibiting proteasomes and inducing apoptosis [15]. In addition, it has been shown, both in vitro and in vivo, that EAAs administration slows the proliferation of CT26 murine colon cancer cells [16].

Although many natural substances are currently in various stages of preclinical and clinical trials due to their potentially selective anticancer effects [17], EAAs supplementation in cancer patients remains highly controversial, despite the frequent need to combat cachexia. This is likely because EAAs, through mTOR, could stimulate tumor proliferation. Furthermore, studies investigating the effects of AAs on cancer have primarily focused on the effects of single AAs or limited groups of AAs, often yielding conflicting results.

To investigate whether EAAs have antiproliferative effects on HCT116 human colon cancer cells, we conducted a preliminary in vitro study to assess the effects of the EAAs mix on inducing the death of HCT116 cells. Subsequently, we performed an in vivo study, using subcutaneous (s.c.) and intraperitoneal (i.p.) injection of HCT116 cells in immunosuppressed athymic nude mice to evaluate differences in tumor volume, adipose, and muscle mass between animals fed a standard laboratory diet (StD) and those fed a special EAA-rich diet (EAARD) as the sole nitrogen source. The aim of this preliminary study is to mainly evaluate the macroscopic evidence related to tumor growth rather than to investigate in detail the molecular mechanisms involved.

## 2. Results

### 2.1. In Vitro Experiments

The neutral red assay results indicate that the EAA mix significantly reduces HCT116 cell viability in both a dose- and time-dependent manner (Figure 1). Notably, after 48 h, a 0.5% concentration of EAAs induces cell death in approximately 10% of the cells, whereas NEAAs tend to promote cell proliferation. Similarly, after 72 h, the presence of EAAs significantly increases cell death by approximately 30% (t = 5.159; *p* = 0.000). Conversely, NEAAs continue to stimulate proliferation (Figure 1A). At a 1% concentration, EAAs induce death in approximately 23% of tumor cells after 48 h and over 53% after 72 h (t = 6.199; *p* = 0.000). In contrast, the mortality rate observed in HCT116 cells with NEAAs remains at approximately 8% at both 48 and 72 h (Figure 1B).

We subsequently assessed the ability of the EAAs mixture to induce apoptosis in cancer cells. Cultured HCT116 cells were exposed to a 1% EAAs mix, and apoptosis was evaluated using Fluorescence-activated cell sorting (FACS) analysis over a 48-h and 72-h time course. The results demonstrated that the EAAs mix significantly increased the number of apoptotic cells after 48 h, with an even greater increase observed after 72 h (Figure 2).

In addition, we evaluated the ability of the EAAs mix to induce autophagy by using immunofluorescence staining for LC3β (Figure 3A–C). Untreated HCT116 cells exhibited very faint LC3β staining, indicating low autophagic activity (Figure 3A,D). In contrast, cells treated with the EAAs mix showed the presence of LC3β-positive autophagosomes starting at a 0.5% concentration after 24 h, with similar observations after 48 h (Figure 3B,E). The intensity of the immunofluorescence staining increased at a 1% concentration, both after 24 h and even more after 48 h, indicating enhanced autophagic activity (Figure 3C,F).

### 2.2. In Vivo Experiments

#### 2.2.1. Sub-Cutaneous (s.c.) HCT116 Injection

The body weight (bw) of both groups of nude mice—those on the StD and those on the EAARD—did not exhibit significant variations throughout the observation period (Figure 4A). No signs of distress were observed in either group. Notably, all mice in the EAARD group completed the treatment period, whereas two mice in the StD group died during treatment. Additionally, it is interesting to note that only one mouse on the StD did not develop any tumors, compared to three mice on the EAARD that remained tumor-free (Figure 4B).

We observed significant differences in the ratio of brown adipose tissue weight (BAT) to bw between the diet groups, with the StD showing a very lower BAT/bw ratio (t = 4.374, *p*-value = 0.000) than EAARD, while no significant differences were noted for retroperitoneal white adipose tissue (rpWAT) (Figure 5A). Additionally, animals fed the EAARD demonstrated the maintenance of muscle mass, specifically in the triceps surae, compared to those fed the StD (t = 16.072; *p*-value = 0.000) (Figure 5B).

Mice fed the EAARD diet exhibited a significant slowdown in s.c. tumor development. By the end of the treatment period, the tumor mass and volume in StD-fed mice were approximately double those in EAARD-fed mice. Specifically, the difference in tumor weight was statistically lower in the EAARD-fed group (t = 26.117; *p*-value = 0.018) (Figure 6A,B and Figure 7A,B). Similarly, the difference in tumor volume was also significantly lower in the EAARD-fed group (t = 22.048; *p*-value = 0.05) (Figure 6C,D and Figure 7F,G).

In StD-fed mice, the tumors developed a superficial intricate vascular network, which was notably reduced in the EAARD-fed group (Figure 7A,B,F,G). After E/H staining, the StD-fed samples revealed a morphologically homogeneous cell population. In contrast, the samples from EAARD-fed mice displayed a quite heterogeneous cell population, featuring numerous nuclei with chromatin condensation and frequent apoptotic cell debris (Figure 7C,H).

Under toluidine blue staining of resin-embedded tumor slides (0.5 µm thick), StD-fed mice exhibited distinct morphological characteristics. At the periphery of the tumor mass, cells appeared morphologically homogeneous, with dark nuclei containing highly condensed chromatin (indicated by arrows in Figure 7D). In the innermost part of the tumor (Figure 7E), the cells displayed heterogeneous shapes and sizes, often exhibiting considerable dimensions. These large tumor cells had abundant and pale cytoplasm, condensed chromatin, and prominent nucleoli. Occasionally, chromatin appeared fragmented, resembling some phases of mitotic division (arrows in Figure 7E). There was no evidence of glandular or squamous morphology.

In contrast, tumors from EAARD-fed mice (Figure 7I,J) demonstrated different morphological features. Peripheral cells (Figure 7I) showed heterogeneous morphology with less organization, and the nuclear staining was pale with condensed chromatin present. In the internal part of the tumor (Figure 7J), cells displayed morphological characteristics like those observed in StD-fed mice; however, they maintained a disorganized architecture with the presence of cellular debris (indicated by arrows). Mitotic cells were only occasionally identified.

To assess apoptosis rates, we conducted staining for activated caspase-3. Tumor cells from StD-fed animals exhibited faint to moderate staining, with occasional cells showing intense immunostaining. In contrast, a significant number of tumor cells from EAARD-fed mice demonstrated intense immunostaining (t = 16.637; *p*-value = 0.000), as shown in Figure 8A,E.

For anti-GRP78 staining, which is related to the role of the endoplasmic reticulum (ER) in protein folding and its involvement in cellular stress responses, we observed that tumor cells in StD-fed animals had faint and occasionally moderate staining. Conversely, in EAARD-fed mice, many cells showed moderate to intense staining (t = 8.448; *p*-value = 0.000), as depicted in Figure 8B,F.

Regarding anti-iNOS staining, an indicator of inflammatory states, intense staining was observed in all cancer cells from StD-fed mice. In contrast, EAARD-fed mice displayed significantly reduced staining, characterized by diffuse faint staining and some cells showing intense staining (t = 11.397; *p*-value = 0.000), as illustrated in Figure 8C,G.

To evaluate the extent of vascularization within subcutaneous tumors, anti-CD31 staining was employed to measure the number of stained vessels per unit area (Vascular density = Vessels/1000 μm^2^). The tumor surface analyzed in each group was approximately 153,000 μm^2^. Vascular density was higher in StD-fed animals compared to EAARD-fed animals (t = 3.964; *p*-value = 0.000), as shown in Figure 8D,H.

#### 2.2.2. Intraperitoneal (i.p.) HCT116 Injection

In our evaluation of the impact of dietary regimens on cancer cell inoculation methods, we performed i.p. injections on two groups of twelve animals each, fed with a StD and an experimental anti-aging and disease-resistant diet (EAARD), respectively. This study aimed to assess the total mass and volume of tumors in the abdomen, which were distinguishable even to the naked eye.

In the StD-fed group, eight mice reached the predetermined endpoint of the trial at 28 days. However, three mice succumbed on days 10, 12, and 17 post-inoculation, and one mouse was euthanized prior to the endpoint due to severe signs of distress caused by cancer, following veterinary advice. In contrast, all mice in the EAARD-fed group successfully reached the end of the treatment period.

Throughout the observation period, no significant differences in body weight (bw) were noted between the StD- and EAARD-fed mice (Figure 9A). However, significant differences were observed in the ratio of certain organ weights to body weight (Figure 9B–F). Notably, StD-fed animals exhibited increased spleen volume and a highly significant reduction in the triceps surae muscle, rpWAT, and BAT. No differences were observed in liver mass between the two dietary groups.

Comparing StD-fed and EAARD-fed groups, significant differences were observed in the number and size of mesenteric tumors (Figure 10). StD-fed animals exhibited a higher number of tumor masses and/or swollen lymph nodes, often presenting a cluster-like appearance. These were found both adjacent to the small intestinal wall and at the mesentery’s root. Conversely, EAARD-fed animals showed reduced or non-visible tumors and swollen lymph nodes (Figure 10 left, A and B).

Both dietary groups included some mice with large, single tumors located on the abdomen’s posterior wall. However, these tumors were significantly smaller in EAARD-fed mice compared to those in StD-fed mice (Figure 10 left, C and D). Overall, the total tumor mass and volume were substantially lower in EAARD-fed animals compared to StD-fed animals (Figure 10 right, A and C). The mean tumor mass and volume also showed significant reductions in the EAARD-fed group (t = 2.663; *p* = 0.015) (Figure 10 right, B and D).

## 3. Discussion

The primary finding from this study is that a specific formulation of free EAAs availability can inhibit human colon cancer growth by promoting cell death through apoptosis and autophagy (AUT) while simultaneously maintaining muscle mass. In cellular systems, programmed cell death is regulated by genes to preserve intracellular stability [18]. However, significant changes in the microenvironment can trigger signal transduction pathways leading to cell death. Among the various types of programmed cell death, AUT-dependent death and apoptosis are crucial and could play a pivotal role in cancer therapy and regulation [19]. These processes are governed by interconnected metabolic molecular mechanisms [20]. Although apoptosis and AUT help maintain homeostasis and regulate cell fate, their interaction can enhance cell death [21]. Thus, modulating these pathways with specific compounds or molecules is beneficial in developing anti-tumor agents [18].

Many anticancer therapies induce tumor cell death by activating caspase-3, a marker of treatment efficacy. Caspases are a conserved family of cysteine proteases involved in cell death and inflammation [22]. However, caspase-3 can also promote stress-induced cancer growth and tumor progression [23,24]. In vitro, we demonstrated that the EAAs mix induces AUT, apoptosis, and cell death within 48 h. This effect was confirmed in EAARD-fed mice, where increased activated caspase-3 levels were observed in subcutaneous tumors, indicating reduced tumor proliferative capacity and decreased neovascularization.

These findings align with previous studies demonstrating the efficacy of the EAAs mix in activating apoptotic pathways in cancer cells without harming non-cancerous cells [15,16]. In cancer cells, EAAs mix intake increases the oxidation of branched-chain amino acids and reduces glycolysis and ATP production. Additionally, levels of NEAAs such as glutamate, glycine, aspartate, and alanine—on which tumor cells heavily rely—were decreased, leading to stress pathway activation, mTORC1 inactivation, protein synthesis blockade, and subsequent apoptosis [25]. Given that glutamate and glycine are precursors for glutathione synthesis, their reduction suggests decreased antioxidant defenses in tumor cells, making them more susceptible to reactive oxygen species (ROS).

In s.c. tumors from EAARD-fed animals, we observed an increased expression of GRP78, also known as BiP, which plays a crucial role in regulating ER functions. GRP78 is involved in protein assembly and folding, managing misfolded proteins for degradation, binding Ca^2+^ to the ER, activating ER-stress sensors, and providing anti-apoptotic support. Cancer cells experiencing chronic stress often induce GRP78 expression as an adaptive response [26]. However, the unfolded protein response (UPR) triggered by severe ER-stress can initiate a series of reactions at both ER and mitochondrial levels, potentially leading to apoptosis [27,28]. Additionally, ER-stress can induce autophagy (AUT), which may result in either cell death or survival [29].

Rapidly growing tumor cells often have high glucose metabolism and glycolytic activity, sometimes exceeding the available blood supply, leading to a metabolic shift known as the Warburg effect, where glucose is converted to lactate even in the presence of oxygen [30]. This creates a tumor microenvironment characterized by glucose deficiency, acidosis, and hypoxia, which contributes to the accumulation of under-glycosylated and misfolded proteins [26]. Previous research has shown that the EAA mixture inhibits mTORC1 and activates ATF4, ER-stress, and apoptotic pathways specifically in cancer cells [25]. Similar findings were observed in murine colon cancer cells in EAARD-fed animals, where increased ER-stress and apoptotic death were noted [16].

Consistent with these findings, our study demonstrates that an EAA-rich diet increases ER-stress, along with the induction of AUT and apoptosis in human colon cancer cells. This suggests that an excess of EAAs severely disrupts the intracellular regulatory systems of tumor cells, creating an imbalance between EAAs and NEAAs availability. This imbalance likely contributes to the impairment of cellular homeostasis and enhances the susceptibility of cancer cells to programmed cell death.

In previous research involving non-cancerous mice, it was demonstrated that an EAARD reduces fat mass by increasing energy expenditure, activating BAT thermogenesis, improving mitochondrial function, promoting metabolic flexibility, and increasing muscle mass [31]. Similar outcomes were observed in colon tumor-bearing mice [16], as well as in the present study, reinforcing these findings.

One of the most significant observations from mice fed with EAARD, especially those with i.p. tumors, is the maintenance of lean mass. Sarcopenia and cachexia, characterized by muscle mass loss, affect over half of cancer patients and directly contribute to about 20% of cancer-related deaths [13,32]. Furthermore, malnutrition, affecting 30 to 90% of cancer patients, can be exacerbated by treatment side effects, impacting quality of life [33]. Mixtures containing all EAAs in stoichiometric ratios have been shown to counteract sarcopenia in both experimental and clinical settings [34,35,36,37]. This is achieved by stimulating protein synthesis through mTORC1 activation, enhancing mitochondrial activity, and regulating oxidative stress [34]. Our study indicates that EAARD helps counteract cancer-associated hyper catabolism, creating an environment unfavorable for tumor cells but not for non-cancerous cells. This suggests a potential nutritional strategy to improve cancer patients’ quality of life and their tolerance to standard therapies.

Tumor cells primarily rely on NEAAs for survival and proliferation [12,25]. Our in vitro experiments confirm that tumor cells can proliferate with NEAAs alone. However, the EAAs mix and EAARD used in this study include serine, an NEAA important for the folate and methionine cycles and glutathione synthesis [38]. Epidemiological data show that folate-enriched diets significantly reduce colon cancer incidence [39]. Serine, along with glutamine and glycine, supports anabolic pathways crucial for cancer cell growth [40]. However, when combined with EAAs, serine acts as an allosteric activator of pyruvate kinase, enhancing glucose oxidation in mitochondria and increasing ROS production, which promotes autophagy [15,41,42]. Our results demonstrate that despite serine’s presence, the EAAs mix and EAARD slow tumor proliferation, indicating a disruption of tumor metabolism without affecting non-tumor cells.

In this study with human colon tumor cells, we observed a slightly lower reduction in tumor volume compared to previous studies with CT26 murine colon tumor cells, despite using the same protocol [16]. This difference may be attributed to the use of immunosuppressed athymic nude mice, which lack an active immune response. Future research should explore the effects of the EAA mixture on the immune system in tumor-bearing animals.

## 4. Materials and Methods

### 4.1. In Vitro Experiments

*Cell Cultures*. The human HCT116 colon carcinoma cell line was a kind gift from Paola Costelli and Fabio Penna (University of Turin, Torino, Italy). The HCT116 cells were cultured in a humidified incubator at 37 °C with 5% CO_2_. The cells were maintained in high-glucose DMEM (Euroclone, Milan, Italy) supplemented with 100 mg/mL penicillin/streptomycin (Sigma Aldrich, Milan, Italy) and 10% FBS (Euroclone, Milan, Italy). Cells were treated with an EAA mix (0.5% and 1%) or an NEAA mix (0.5% and 1%) (Table 1) dissolved in complete medium (DMEM). Control cells were treated with DMEM alone.

Neutral Red Assay. Cell viability was measured by neutral red assay. Cells (2 × 10^3^) were seeded in triplicate in 96-well plates and left to grow in complete medium for 24 h. Then, cells were incubated with a medium supplemented with EAAs mix or NEAAs mix at 0.5 or 1% concentration. After 48 and 72 h, medium was replaced with DMEM supplemented with 5% FBS and 40 μg/mL neutral red dye, and plates were incubated at 37 °C for 2 h. Then, cells were PBS-washed and incubated with a de-staining solution (50% ethanol in deionized water with 1% acetic glacial acid). Plates were shaken until complete dye extraction was achieved, and then absorbance was measured by reading the plate at 540 nm emission wavelengths.

Flow Cytometric Analysis. Cell apoptosis was assessed using an Annexin V/Propidium Iodide (PI) apoptosis-detection kit (Immunostep Biotec, Salamanca, Spain), according to the manufacturer’s instructions. Cells (5 × 10^4^) were seeded duplicated into six-well plates and left to grow in complete medium for 24 h. Then, cells were incubated with a medium supplemented with EAAs at 1% concentration. After 48 and 72 h, cells were collected into flow cytometry tubes, PBS washed, resuspended in binding buffer, and double stained with Annexin-V-FITC/PI. Doxorubicin was used as a positive control. Cytofluorimetric analysis was performed using a MACSQuant Analyzer (Miltenyi Biotec, Bologna, Italy). Cell debris, doublets, and aggregates were excluded from the analysis, and 20,000 events per sample were analyzed.

Immunofluorescence analysis. Cells were cultured onto 12 mm glass coverslips in 24-well plates (2 × 10^4^) and left to grow in complete medium for 24 h. Then, cells were incubated with a medium supplemented with EAAs mix or NEAAs at 0.5 or 1% concentration. After 24 and 48 h, cells were fixed with paraformaldehyde (PFA) solution (3% PFA in PBS) for 20 min at 4 °C, permeabilized with 0.1% Triton X-100 in PBS for 10 min at room temperature (RT), and then blocked with a bovine serum albumin (BSA) solution (1% BSA in PBS with 0.1% sodium azide) for 30 min at RT. Cells were incubated with primary antibody anti-LC3β (code: sc-376404; Santa Cruz Biotechnology, Dallas, TX, USA) for 3 h at RT, washed, and then incubated with secondary antibody for 45 min at RT, protected from light. Nuclei were counterstained with Hoechst Nucleic Acid Stains dye (cod. H1399, ThermoFisher Scientific Inc., Waltham, MA, USA) for 30 s at RT and samples were mounted on slides using Mowiol^®^ 4-88 (code 475904, Merck, Darmstadt, Germany) mounting media. Images were acquired by using a fluorescent Axio observer microscope equipped with Apotome (Carl Zeiss, Oberkochen, Germany) using Zen 3.5 (Blue Version) software (Carl Zeiss, Oberkochen, Germany).

### 4.2. In Vivo Experiments

The experimental protocol was approved and conducted in accordance with laws of the Italian Ministry of Health and complied with the “The National Animal Protection Guidelines”. The Ethical Committee for animal experiments of the University of Brescia (OPBA) and the Italian Ministry of Health had approved the procedures (decree n. 539/2021-PR).

Diets. Standard laboratory rodent food (Mucedola srl, Milan, Italy) was used as the reference standard diet (StD). The special EAAs-rich diet (EAARD) (Dottori Piccioni srl, Gessate, Milano, Italy) matched the same total macronutrients, micronutrients, and calorie contents. The primary difference between diets was the source and type of nitrogen. In StD, the nitrogen source was represented by unspecified vegetal and animal (fish) proteins. As with all food proteins, the EAA-to-NEAA ratio (EAA/NEAA) should be considered < or <<0.9. From the whole protein mix, it is impossible to obtain the exact percentage in EAAs and NEAAs. In contrast, EAAs mix provided nitrogen as free AAs, where the EAAs are in excess (84%) compared to NEAAs (16%) (EAA/NEAA = 6.14), as previously described [43]. In summary, both the EAAs mix and the EAAs diet incorporated L-cystine (a NEAA) to satisfy the requirements for sulfur AAs while keeping methionine content to a minimum. Additionally, serine (a NEAA) was included to optimally sustain the folate and methionine cycle, as well as energy production. The composition of EAARD is summarized in Table 1.

Animals. Fifty BALB/c-nude (athymic) male mice, aged 5 weeks (Envigo srl, San Pietro al Natisone, Udine, Italy), were individually housed in filter cages and kept on a 12/12 h light/dark cycle (lights on from 7am to 7pm) in a quiet, temperature- and humidity-controlled room. After 7 days of ad libitum access to StD and water, the mice were randomized into two groups. The first group (n = 25) continued with the StD, while the second group (n = 25) was switched to the EAARD. Every three days, body weight (bw), food, and water consumption were measured. After 15 days, 13 StD-fed mice and 13 EAARD-fed mice were injected subcutaneously (s.c.) at the right hip with 1 × 10^5^ HCT116 cells (ATCC code CCL-247™) in 100 μL of physiological solution in Matrigel (1:1). An additional 24 mice (12 StD-fed and 12 EAARD-fed) were intraperitoneally (i.p.) injected with 1 × 10^6^ HCT116 cells. Daily monitoring was performed, and, if necessary, animals were euthanized early based on veterinary advice and Ethics Committee Criteria. Twenty-one days after the s.c. injection, the animals were sacrificed, and tumors removed. Retroperitoneal WAT (rpWAT), BAT, and triceps surae were isolated, measured, and weighed. Twenty-eight days after the i.p. injection, the animals were sacrificed. Liver, spleen, rpWAT, BAT, and triceps surae were isolated, measured, and weighed. Visible i.p. tumors with a diameter greater than 1 mm located on the mesentery and/or on the parietal peritoneum were measured and weighed. Tumor volume (mm^3^) was calculated as:Vol = [(max diameter × smallest diameter^2^)/2]

All tumors were adequately prepared for histochemical and immunohistochemical analyses.

Histology and Immunohistochemistry. Tumor samples were immediately removed, quickly washed in physiological solution, fixed overnight in 10% neutral buffered formalin, processed, and embedded in paraffin using an automatic includer (Donatello-2, Diapath s.p.a, Martinengo, BG, Italy). Histopathological analysis was conducted under eosin and hematoxylin (E/H) staining. Sections were obtained from five animals per group. Five sections were obtained from each tumor, spaced approximately 100 μm apart. For immunohistochemistry (IHC), tumor sections (thick about 5 μm) were incubated overnight with primary polyclonal anti-CD31 (28083-1-AP), anti-GRP78 (11587-1-AP), and recombinant anti-iNOS (80517-1-RR), all from Proteintech (Rosemont, IL, USA), and anti-active Caspase-3 (NB100-56113) from Novus Biologicals (Easter Ave Centennial, Centennial, CO, USA). The sections were processed according to the manufacturer’s protocol and visualized with IHC Prep & Detect Kit for Rabbit Primary Antibody (PK10017, Proteintech). The IHC negative control was performed by omitting the primary antibody in the presence of iso-type-matched IgGs. The staining intensity was evaluated using an Olympus BX50 microscope equipped with an image analysis program (Image Pro-Plus 4.5.1, Immagini e Computer, Milano, Italy). The IOD was calculated for arbitrary areas by measuring 30 fields for each sample using a 20× lens. Other tumor samples were fixed with 3% glutaraldehyde in a cacodylate buffer (pH 7.4, 0.2 M) and processed with standard procedures for embedding in Araldite (Sigma-Aldrich Chemical Co., Milan, Italy). Thick sections (about 0.5 µm) were stained with toluidine blue and used for morphometry.

### 4.3. Statistics

Data are expressed as mean ± SD. To compare the data of experimental groups, we used the two-sample Welch *t*-test, which does not require the assumption of equal variance across populations (https://www.statskingdom.com/). The value of *p* < 0.05 was considered statistically significant.

## 5. Conclusions

The literature and our preliminary data suggest that a balanced EAAs mixture does not promote tumor proliferation; rather, it has the opposite effect by inducing apoptosis and autophagy in tumor cells, promoting anabolic stimuli in healthy cells while preserving muscle mass. Therefore, dietary supplementation with EAAs could be an effective adjunct to standard therapies, combating tumor-induced hyper catabolism, preserving muscle mass, and creating an unfavorable environment for tumor growth. This approach could enhance patient quality of life and potentially improve survival outcomes.

Study limitations. This study is mainly based on highlighting the macroscopic effects of EAAs on the proliferation of tumor cells. To this end, we primarily utilized immunohistochemistry and immunofluorescence to support our findings which are substantially evidence-based. We chose these techniques due to the varying differentiation states of the tumor cells. Immunohistochemistry and immunofluorescence allow for a more detailed examination of the physiological and pathological changes and their localization within the tissue, which can be more effectively highlighted than with molecular methods. While molecular analyses offer heightened sensitivity, they involve tissue homogenization, which overlooks the structural organization, cell type, and intracellular localization of the examined molecules [31]. In our view, this constitutes a significant limitation in the exclusive use of molecular analysis. However, we recognize that more comprehensive molecular investigations will be necessary for a better understanding of the metabolic pathways that render tumor cells more sensitive to the microenvironmental changes induced by the EAA mixture. This will help inform the development of more targeted and effective therapeutic approaches for cancer treatment.

## Figures and Tables

**Figure 1 ijms-26-07014-f001:**
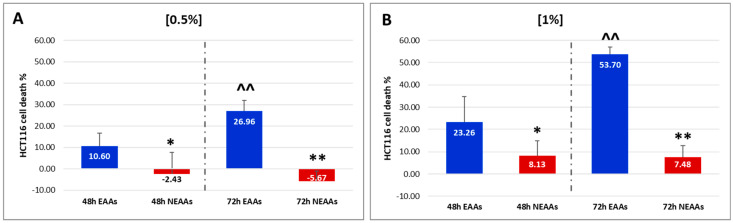
(**A**,**B**) Quantitative values of HCT116 death treated with 0.5% (**A**) and 1% (**B**) of essential amino acids (EAAs—blue columns) and non-essential AAs (NEAAs—red columns). Values are expressed as mean ± sd. 48 h or 72 h, EAAs vs. NEAAs: * *p* < 0.05; ** *p* < 0.01. EAAs or NEAAs, 48 h vs. 72 h: ^^ *p* < 0.01.

**Figure 2 ijms-26-07014-f002:**
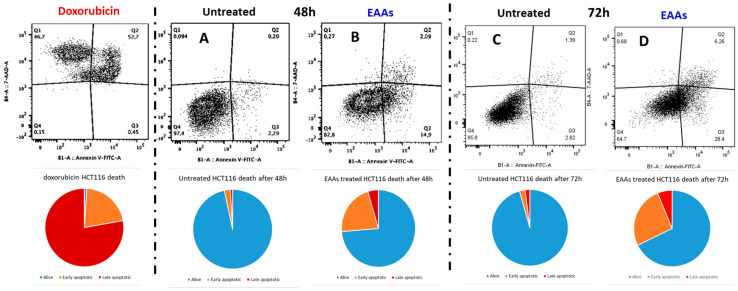
Fluorescence-activated cell sorting (FACS) analysis was conducted to examine apoptosis in HCT116 cells treated with 1% of essential amino acids mixture (EAAs) at 48 h (**A**,**B**) and 72 h (**C**,**D**). Doxorubicin (1 µM for 48 h) was used as positive control. In the pie charts, the red color indicates late apoptotic cells, the orange color early apoptotic cells, and the light blue color alive cells.

**Figure 3 ijms-26-07014-f003:**
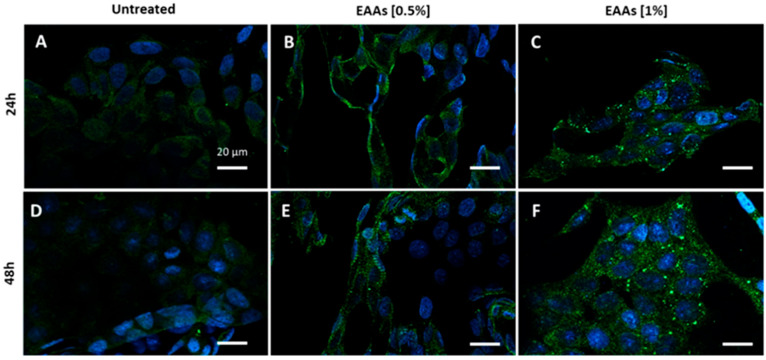
Immunofluorescence for LC3β (green dots) on HCT116 cells incubated without (untreated) or with the essential amino acid mixture (EAAs) at concentrations of 0.5% and 1% for periods of 24 h (**A**–**C**) and 48 h (**D**,**E**). Exposure to the EAAs mix induced the formation of autophagosomes to a greater extent at the 1% concentration already after 24 h (**C**) and even more so after 48 h (**F**). Nuclei are in blue. Scale bar = 20 μm.

**Figure 4 ijms-26-07014-f004:**
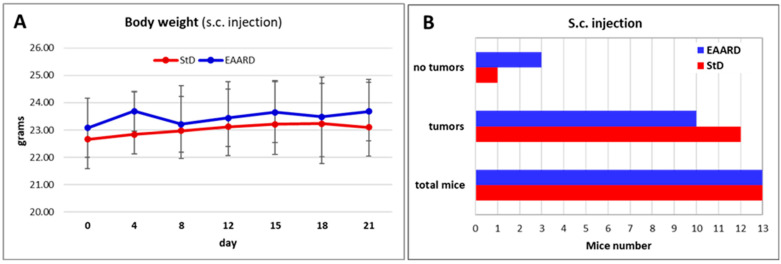
(**A**) Body weight trend during the observation period as a function of diet. There are no significant differences between the two groups (n = 13, for each group). Values are expressed as mean ± sd. (**B**) Graphical representation of comparison between the StD (red) and the essential amino acid-rich diet (EAARD) (blue) regarding the development of the s.c. tumor with respect to the total mice inoculated.

**Figure 5 ijms-26-07014-f005:**
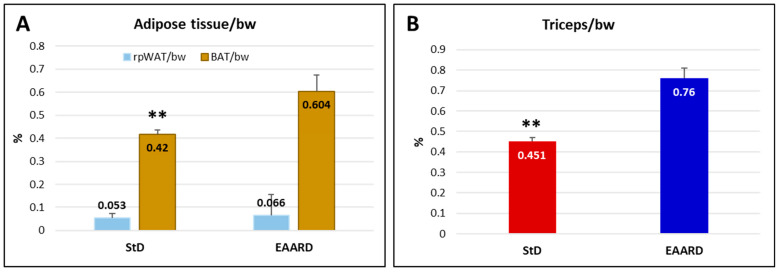
(**A**) Ratio of retroperitoneal white adipose tissue (rpWAT) and brown adipose tissue (BAT) to body weight (bw). In StD-fed animals, a significant decrease in the BAT/bw ratio is observed. (**B**) Triceps surae weights were significantly lower in mice fed with the StD. n = 13, for each group. Values are expressed as mean ± sd. ** *p* < 0.01.

**Figure 6 ijms-26-07014-f006:**
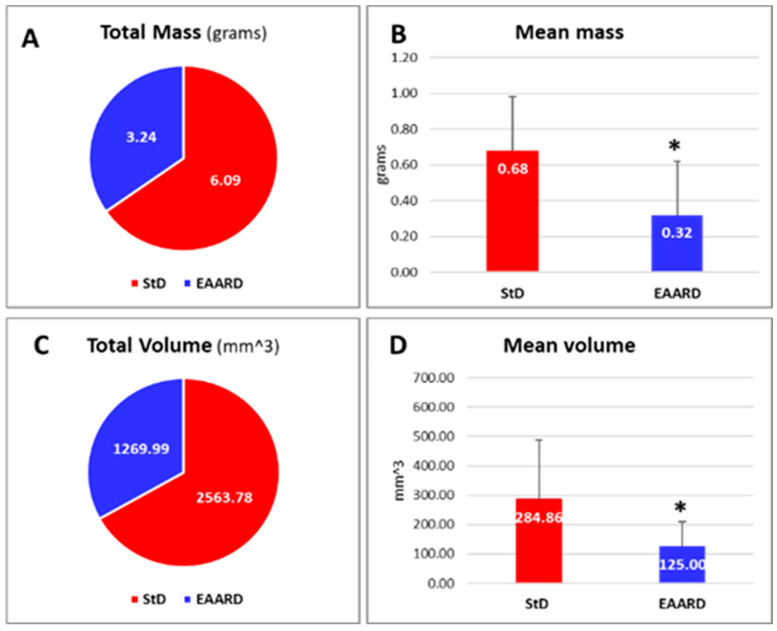
(**A**,**B**) Total and mean wight (mass in grams) of s.c. tumors according to diets. (**C**,**D**) Total and mean volume (mm^3^) of tumors according to diets. n = 13, for each group. Values are expressed as mean ± sd. * *p* < 0.05.

**Figure 7 ijms-26-07014-f007:**
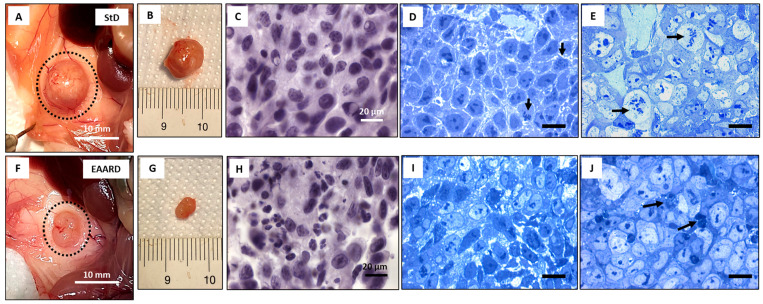
Representative images of sub-cutaneous tumors. (**A**–**E**) StD-fed mice and (**F**–**J**) essential amino acid-rich diet (EAARD)-fed mice. (**A**,**F**) tumor in situ and (**B**,**G**) explanted. Note the notable difference in size and the numerous thin blood vessels present on the tumor surface in StD-fed animals. (**C**,**H**) representative histologic section Eosin/Hematoxylin stained that highlights the different cellular organization of the EAARD-fed tumor. (**D**,**E**,**I**,**J**) Toluidine blue staining at the periphery (**D**,**I**) and at the center of the tumor (**E**,**J**). (**D**,**E**) The nuclei have a dark color with the presence of highly condensed and fragmented chromatin (arrows). (**I**,**J**) the cells show a disorganized architecture with the presence of cellular debris (arrows). Scale bar = 20 µm.

**Figure 8 ijms-26-07014-f008:**
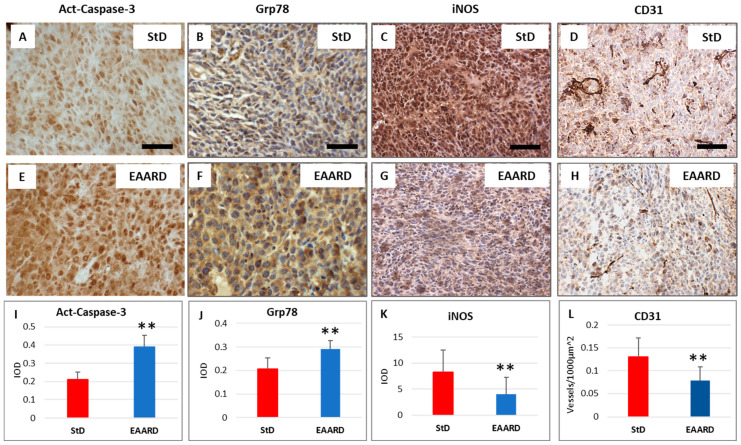
Sub-cutaneous tumors. Representative images of immunohistochemistry in standard diet (StD)-fed (**A**–**D**) and essential amino acid-rich diet (EAARD)-fed mice (**E**–**H**). Scale bar = 20 μm. The graphs indicate the optical density for active-Caspase-3, GRP78, and iNOS (**I**–**K**). The graph for CD31 (**L**) is expressed as Vascular density = Vessels/1000 μm^2^. Values are expressed as mean ± sd. ** *p* < 0.01.

**Figure 9 ijms-26-07014-f009:**
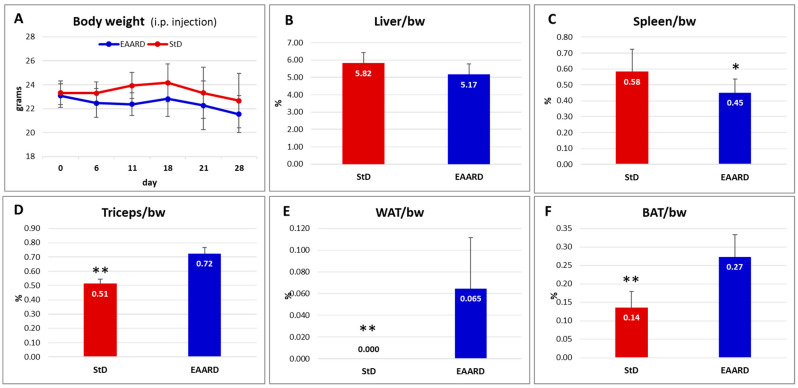
I.p. injection. (**A**) Body weight (bw) trend between the inoculation (day 0) and sacrifice (day 28). Red line = standard diet (StD)-fed mice (n = 12). Blue line = essential amino acids rich diet (EAARD)-fed mice (n = 12). No significant differences in bw were found. (**B**–**F**) Ratio between organ weight [liver, spleen, triceps surae, retroperitoneal white adipose tissue (WAT), and brown adipose tissue (BAT)] and bw according to diet. StD-fed animals (n = 9, red columns) show a significant tendency towards splenomegaly (**C**), as well as a massive reduction in triceps muscle mass (**D**), WAT (**E**), and BAT (**F**). EAARD-fed animals (n = 12) are represented with the blue columns. Values are expressed as mean ± sd. * *p* < 0.05, ** *p* < 0.01.

**Figure 10 ijms-26-07014-f010:**
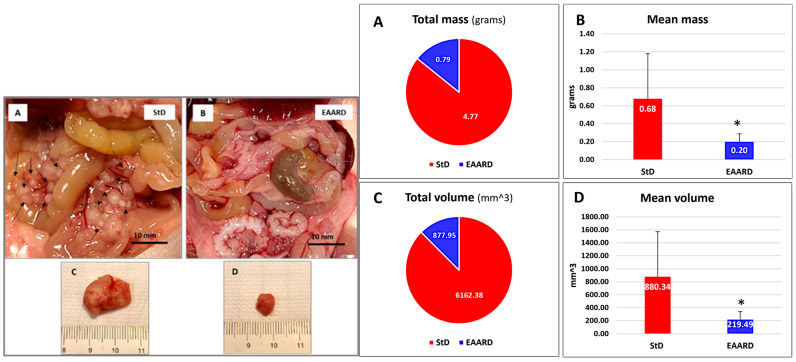
i.p. injection. **Left** (**A**–**D**): abdominal tumors clearly distinguishable even to the naked eye. (**A**,**B**) Representative image of the mesenteric tumor (black arrows) in standard diet (StD)-fed (**A**) and essential amino acid-rich diet (EAARD)-fed (**B**), respectively. Scale bar: 10 mm. (**C**,**D**) Representative image of large intra-abdominal tumors sometimes observed in both groups. Note that in EAARD-fed mice the tumor volume is significantly smaller than in those with StD. **Right** (**A**–**D**): Total and mean mass (**A**,**B**) and volume (**C**,**D**) of tumors according to diets. EAARD-fed animals (n = 12) are represented in blue, and StD-fed (n = 9) in red. Values are expressed as mean ± sd. * *p* < 0.05.

**Table 1 ijms-26-07014-t001:** Composition of EAAs mix and NEAAs mix for in vitro experiments, and composition of pellets (StD and EAARD) for mice. * Nitrogen (%) from free AAs only. ° Nitrogen (%) from vegetable and animal proteins and added AA. StD = standard diet; EAARD = essential-AA-rich diet; N = nitrogen. bcaa = branched chain AA.

	EAAs Mix	NEAAs Mix	StD	EAARD
KCal/Kg	--	--	3952	3995
Carbohydrates (%)	--	--	54.61	61.76
Lipids (%)	--	--	7.5	6.12
Nitrogen (%)	--	--	21.8 °	20 *
Proteins: % of total N content	--	--	95.93	--
Free AA: % of total N content	--	--	4.07	100
EAA/NEAA (% in grams)	--	--	< or <<0.9	6.14 (86/14)
Free AA composition				
L-Leucine (bcaa)	13.53%	--	--	13.53%
L-Isoleucine (bcaa)	9.65%	--	--	9.65%
L-Valine (bcaa)	9.65%	--	--	9.65%
L-Lysine	11.6%	--	0.97%	11.6%
L-Threonine	8.7%	--	--	8.7%
L-Histidine	11.6%	--	--	11.6%
L-Phenylalanine	7.73%	--	--	7.73%
L-Methionine	4.35%	--	0.45%	4.35%
L-Tyrosine	5.80%	1.0%	--	5.80%
L-Tryptophan	3.38%	--	0.28%	3.38%
L-Cystine/Cysteine	8.20%	--	0.39%	8.20%
L-Alanine	--	35.0%	--	--
L-Glycine	--	15.0%	0.88%	--
L- Arginine	--	14.0%	1.1%	--
L-Proline	--	12.0%	--	--
L-Glutamine	--	12.0%	--	--
L-Serine	2.42%	6.0%	--	2.42%
L-Glutamic Acid	--	2.0%	--	--
L-Asparagine	--	2.0%	--	--
L-Aspartic Acid	--	1.0%	--	--
Ornithine-αKG	2.42%	--	--	2.42%
N-acetylcysteine	0.97%	--	--	0.97%

## Data Availability

The data are available upon motivated request to the corresponding author.

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
