# Peer review of "A Diet Rich in Essential Amino Acids Inhibits the Growth of HCT116 Human Colon Cancer Cell In Vitro and In Vivo"

_ijms, 2025, doi:10.3390/ijms26147014_

Round 1
Reviewer 1 Report
Comments and Suggestions for Authors
In the following work, Corsetti et al. evaluate how a mixture of essential amino acids (EAAs) are able to prevent growth of HCT116 cells by inducing autophagy and apoptosis, increasing endoplasmic reticulum stress, and inhibiting inflammation and neo-vascularization. Their experiments include both in vitro and in vivo models.
Major points
- In the in vitro experiments, Figure 1, the authors should add a control condition (normal cell growth medium) to this experiment.
Furthermore, it would strengthen the results if the authors measured changes in cell proliferation, for example, by MTT or similar.
- In in vivo experiments, how would the increase in brown adipose tissue be explained? Several recent studies demonstrate how increased brown adipose tissue correlates with more pronounced tumor development. The authors should discuss this.
- In the Discussion section the authors write: “This effect was confirmed in EAARD-fed mice, where increased activated caspase-3 levels were observed in subcutaneous tumors, indicating reduced tumor proliferative capacity and decreased neovascularization.” I believe that measuring caspase 3 alone does not allow the authors to conclude the relevance of apoptosis regulation. They should measure some other mediator involved in the apoptosis process.
- Regarding the results presented in Figure 7, the authors should explain this result. Why would a disorganized histoarchitecture, along with heterogeneous cellular morphology, imply a benefit?
Minor points
In Figure 9, the authors must arrange the figures because they are not as "left" and "right".
Author Response
We sincerely thank the referee for his time and valuable suggestions. We trust that the responses and the changes made based on his suggestions are comprehensive and have improved the quality of the work.
Major points
[Comment/question]. In the in vitro experiments, Figure 1, the authors should add a control condition (normal cell growth medium) to this experiment.
[Answer] In the graph in figure 1, the control is not present as it represents the % mortality of the cells treated with EAA and NEAA compared to the control cells (treated with the normal cell growth medium), which have zero mortality.
[Comment/question]. Furthermore, it would strengthen the results if the authors measured changes in cell proliferation, for example, by MTT or similar.
[Answer] As for the proliferation of tumor cells with EAA, we did not measure it precisely because we had observed cell death.
[Comment/question]. In in vivo experiments, how would the increase in brown adipose tissue be explained? Several recent studies demonstrate how increased brown adipose tissue correlates with more pronounced tumor development. The authors should discuss this.
[Answer] Thank you for your comment, and I realize that the result, as written, is confusing, so I apologize. In reality, it isn't an increase in BAT but rather a maintenance of its mass, while a significant reduction is observed in the StD group. I have amended the text and Figure 5 accordingly.
[Comment/question]. In the Discussion section the authors write: “This effect was confirmed in EAARD-fed mice, where increased activated caspase-3 levels were observed in subcutaneous tumors, indicating reduced tumor proliferative capacity and decreased neovascularization.” I believe that measuring caspase 3 alone does not allow the authors to conclude the relevance of apoptosis regulation. They should measure some other mediator involved in the apoptosis process.
[Answer] I thank you and agree with your observation. However, the primary purpose of the in vivo study is to investigate the macroscopic morphological evidence of changes in tumor growth with a diet rich in EAA. While it is true that measuring caspase-3 alone is not a reliable indicator of apoptosis, the evident reduction in tumor mass, combined with the increased expression of caspase-3, legitimizes the hypothesis that apoptosis may play a role in containing tumor proliferation, as also demonstrated in in vitro experiments. Further studies are certainly needed to investigate in detail and definitively the mechanisms underlying the inhibition of tumor proliferation and/or death induced by EAA.
[Comment/question]. Regarding the results presented in Figure 7, the authors should explain this result. Why would a disorganized histoarchitecture, along with heterogeneous cellular morphology, imply a benefit?
[Answer] Disorganized cytoarchitecture is present in tumor tissue from animals fed EAARD (Fig. 7 I, L). This does not represent an advantage for the tumor tissue, but rather indicates its morpho-functional impairment, consistent with its growth difficulty.
Minor points
[Comment/question]. In Figure 9, the authors must arrange the figures because they are not as "left" and "right".
[Answer] As suggested, we have changed the numbering of the figures.
Reviewer 2 Report
Comments and Suggestions for Authors
Paper titled (nhibitory effects of special mixture of amino acids on human 2 colon cancer cell (HCT116) in vitro and in vivo. ) by Giovanni Corsetti and his colleagues described the inhibitor effect of an aminoacid mixture on the growth of colon cancer cells grown in vitro as well as if grown in vivo. This is a very humble trial for tumor inhibitory activity and the authors provided only descriptive results without mechanistic approach on the claimed activity
Also the stat analysis seems to be done inccurately as most of data seem to be non parametric as indicated by the high SD value
1- My first commnet is on the title: (i nhibitory effects of special mixture of amino acids on human colon cancer cell (HCT116) in vitro and in vivo ) which is not informative enough about confirming whether there was an inhibitory effect at the end or not and what was the mechanism by which this inhibition was mediated (eg inhibitng angiogenesis, promoting apoptosis...etc)
Also we can say (HCT116 human colon cancer cell) instead of the way presented in the title
please indicate that the amino acides are given by diet
2- Abstract of the paper is short and must be amended by some numerical values from the key results
3 - Key words: ( cancer; amino acids; colon; human; diet; mice ) is can be amended, please use the best words for indexing purposes. please add the name of the amino acids if possbile and combine (cancer + colon + human) to be (human colon cancer) and add a key word for the (HCT116 cells), please improve the key words
4 - The intro is long and must be shortened to be more concrete and explore the novelty and research gap for this study
5 - The last paragraph in the intro representing the aim of the study needs to be rewritten as it looks like a conclusion, it should be a clear aim describing the goal of the study and how you planned to acheive it
6 - Some long parts in the into comes with a single refernces. Please add and cite appropriate refernces
7 - Images in figure 8 are captured at different microscopic powers, please provide equal powers for appropriate comprisons
8 - what was the weight of mice at the begin of the experiment
9 - a diagram showing the step wise actions and prpearation for the behavior task is a worth addition to this paper
10 - In methods , Mention in details the housing conditions
11 - how authors were keen to reduce animal suffering
12 - Mention the methods of anesthesia and the type of anesthetic in details
13 - Authors should mention whether any data sets did not follow normal dist after Shapiro Wilk test
14 - Use appropriate abbreviations for minutes, seconds...etc
15 - Authors should give the source of chemicals, kits and antibodies completely and consistently (code, company, town, state and country) & version for software
16 - Every abbreviation in figures should be explained in the figure legend to be self explanatory & stands alone.
17 - Mention "n" in each illustration individually
18- please mention why amino acids inhibited tumor growth in abstrat
19- please discuss thoroughly why amino acids inhibited tumor growth in discussion
20 - please add the limitation and future directions of this study at the end of the conclusion
21- for histology stains: how many sections were taken from each animal and how many animals from each group?
With the above revisions, I believe your manuscript will make a valuable contribution to the field. I encourage you to address these suggestions to improve the clarity and overall impact of your paper.
Le
Author Response
We sincerely thank the referee for his time and valuable suggestions. We trust that the responses and the changes made based on his suggestions are comprehensive and have improved the quality of the work.
Paper titled (inhibitory effects of special mixture of amino acids on human 2 colon cancer cell (HCT116) in vitro and in vivo. ) by Giovanni Corsetti and his colleagues described the inhibitor effect of an aminoacid mixture on the growth of colon cancer cells grown in vitro as well as if grown in vivo. This is a very humble trial for tumor inhibitory activity and the authors provided only descriptive results without mechanistic approach on the claimed activity. Also the stat analysis seems to be done inccurately as most of data seem to be non parametric as indicated by the high SD value.
[Answer] We thank you for your detailed observation. Indeed, this study is primarily based on the description of evidence-based morphological findings regarding the effect of a diet rich in EAAs on tumor development. We considered these preliminary data of potential interest, because they confirm the adverse effect of EAAs on the progression of human tumor cells, similar to what was previously observed for murine tumor cells. For this reason, as stated in the introduction (line 100-102), and based on available resources, we did not delve into the mechanistic aspect, as stated within the limits of the study. This aspect will certainly need to be investigated with further studies. Since the data collected do not follow a normal distribution, we used a non-parametric test (see statistics).
1- My first commnet is on the title: (inhibitory effects of special mixture of amino acids on human colon cancer cell (HCT116) in vitro and in vivo ) which is not informative enough about confirming whether there was an inhibitory effect at the end or not and what was the mechanism by which this inhibition was mediated (eg inhibitng angiogenesis, promoting apoptosis...etc). Also we can say (HCT116 human colon cancer cell) instead of the way presented in the title please indicate that the amino acides are given by diet.
[Answer] We changed the title.
2- Abstract of the paper is short and must be amended by some numerical values from the key results
[Answer] we have modified the abstract by inserting the most striking data.
3 - Key words: ( cancer; amino acids; colon; human; diet; mice ) is can be amended, please use the best words for indexing purposes. please add the name of the amino acids if possbile and combine (cancer + colon + human) to be (human colon cancer) and add a key word for the (HCT116 cells), please improve the key words
[Answer] We believe we've improved the keywords. It's not possible to include the names of all the AAs in the blend, so we've limited ourselves to "essential amino acids.".
4 - The intro is long and must be shortened to be more concrete and explore the novelty and research gap for this study
[Answer] We have shortened the introduction to better explain the novelty and potential usefulness of our experimental approach.
5 - The last paragraph in the intro representing the aim of the study needs to be rewritten as it looks like a conclusion, it should be a clear aim describing the goal of the study and how you planned to acheive it
[Answer] We deleted the last paragraph.
6 - Some long parts in the into comes with a single refernces. Please add and cite appropriate refernces
[Answer] We have modified the introduction.
7 - Images in figure 8 are captured at different microscopic powers, please provide equal powers for appropriate comprisons
[Answer] We replaced photos 8A and 8E, making the magnification the same as the other photos.
8 - what was the weight of mice at the begin of the experiment
[Answer] In the subcutaneous injection group, the initial weight was 22.66 ±1.51 (StD) and 23.07 ±1.08 (EAARD) as indicated in Figure 4A. In the intraperitoneal injection group, the initial weight was 23.33 ±1.01 (StD) and 23.08 ±0.98 (EAARD) as indicated in Figure 9A.
9 - a diagram showing the step wise actions and prpearation for the behavior task is a worth addition to this paper
[Answer] I apologize, but I don't understand what's required. We've summarized the protocol's main steps and results in the graphical abstract.
10 - In methods , Mention in details the housing conditions
[Answer] Housing conditions are described in Materials and Methods, under Animals, Lines, etc. We have specified some characteristics of the housing room. If you deem other parameters necessary, please let us know.
11 - how authors were keen to reduce animal suffering
[Answer] In vivo experiments, the standards set by national and international laws regarding animal protection and welfare for scientific purposes were applied (see line 415-418). In particular, after inoculation of tumor cells, the animals were assessed daily by the responsible veterinarian to determine behavioral/postural changes indicative of suffering, as stated in the Materials and Methods, Animals section (line 443-444).
12 - Mention the methods of anesthesia and the type of anesthetic in details
[Answer] Subcutaneous and intraperitoneal injections do not require anesthesia. The animals were sacrificed by cervical dislocation, which does not require anesthesia.
13 - Authors should mention whether any data sets did not follow normal dist after Shapiro Wilk test
[Answer] Since the Shapiro-Wilk test showed a significant departure from normality, we used the Welch t-test, which does not require the assumption of equal variance across populations.
14 - Use appropriate abbreviations for minutes, seconds...etc
[Answer] We have checked the text for correct abbreviations of the units of measurement used.
15 - Authors should give the source of chemicals, kits and antibodies completely and consistently (code, company, town, state and country) & version for software
[Answer] The required information regarding the main chemicals used and the antibody codes are described in the Materials and Methods sections of the text. We have added the missing ones.
16 - Every abbreviation in figures should be explained in the figure legend to be self explanatory & stands alone.
[Answer] We have explained the abbreviations in the figure captions.
17 - Mention "n" in each illustration individually
[Answer] In the figures legend we have mentioned "n".
18- please mention why amino acids inhibited tumor growth in abstrat
[Answer] We have specified why AAs prevent the proliferation of tumor cells.
19- please discuss thoroughly why amino acids inhibited tumor growth in discussion
[Answer] We would be happy to describe in detail the mechanism by which this AA mixture inhibits tumor growth. However, as stated in the text and in a previous comment, the data collected so far are based on morphological evidence and the variation of certain immunohistochemical parameters, which certainly do not allow for a comprehensive and definitive explanation of the mechanism of tumor metabolic inhibition, which is likely very complex at the molecular level. As we have discussed, we can only highlight some parallels and confirmations with previous publications, which are very few that use complete EAA mixtures like ours, and propose hypotheses. The aspect we believe to be important with the available data, albeit on an experimental model but with human cells, consists in proposing a paradigm shift in the nutritional approach to cancer patients, where complete and balanced amino acid supplementation should no longer be seen as a dangerous tumor promoter, but as a tool that improves the patient's quality of life by maintaining muscle mass and the metabolism of healthy cells, instead creating an environment unfavorable to the development of tumor cells.
20 - please add the limitation and future directions of this study at the end of the conclusion
[Answer] we moved the limitations after the conclusions.
21- for histology stains: how many sections were taken from each animal and how many animals from each group?
[Answer] Sections were obtained from 5 animals per group. Five sections were obtained from each tumor, spaced approximately 100 μm apart.
Reviewer 3 Report
Comments and Suggestions for Authors
The manuscript entitled “Inhibitory effects of special mixture of amino acids on human colon cancer cell (HCT116) in vitro and in vivo” provides a valuable data on a socially significant problem, such as the search for strategies for improvement of cancer patients' quality of life and treatment options. The manuscript is well-written and structured. The methods are precisely described and include studies on both, cell cultures and experimental animals. The results are very well presented and clearly explained.
However, some typographical errors could be found in the text, for example: “in vivo” and “in vitro” should be in Italic; “(StD” should be corrected to” (StD)”; “CO2” – 2 should be in a subscript.
I would recommend some of the keywords to be revised to more specific to the research.
In my opinion, the sentence “To date, no alternative therapeutic methods, or complementary treatments to standard therapies, have been able to save patients' lives or alleviate disease-related pain, thereby improving their quality of life.” should be revised based on the fact that at present a lot of natural-derived substances are at different stages of pre-clinical and clinical studies due to the selective anticancer effect they could exhibit.
Author Response
We sincerely thank the referee for his time, appreciation, and valuable suggestions.
[comment] some typographical errors could be found in the text, for example: “in vivo” and “in vitro” should be in Italic; “(StD” should be corrected to” (StD)”; “CO2” – 2 should be in a subscript.
[Answer] we have made the suggested corrections.
I would recommend some of the keywords to be revised to more specific to the research.
[Answer] we revised the keywords
In my opinion, the sentence “To date, no alternative therapeutic methods, or complementary treatments to standard therapies, have been able to save patients' lives or alleviate disease-related pain, thereby improving their quality of life.” should be revised based on the fact that at present a lot of natural-derived substances are at different stages of pre-clinical and clinical studies due to the selective anticancer effect they could exhibit.
[Answer] we changed the sentence.
Round 2
Reviewer 1 Report
Comments and Suggestions for Authors
The authors have modified the manuscript and responded to the comments/suggestions made appropriately.
Author Response
We thank this reviewer for his time and for his suggestions that allowed us to improve the text.
Reviewer 2 Report
Comments and Suggestions for Authors
The revised version of paper titled (Inhibitory effects of special mixture of amino acids on human colon cancer cell (HCT116) in vitro and in vivo. ) by Authors Giovanni Corsetti
et al. was partly revised and suthors should mention every change made in the resposnses. Authors should mention were it was elaborated in the text (page and line) to follow these changes inside the manuscript, NOt jut found in the reply to reviewers
Best regards
Author Response
We thank the reviewer for the clarification. We apologize if we did not clearly indicate the location of the changes made in the reply to the referee. We believed it appropriate and sufficient to highlight the modified text in yellow. Below, we provide the page and line indications of the changes.
1- My first commnet is on the title: (inhibitory effects of special mixture of amino acids on human colon cancer cell (HCT116) in vitro and in vivo ) which is not informative enough about confirming whether there was an inhibitory effect at the end or not and what was the mechanism by which this inhibition was mediated (eg inhibitng angiogenesis, promoting apoptosis...etc). Also we can say (HCT116 human colon cancer cell) instead of the way presented in the title please indicate that the amino acides are given by diet.
[Answer 1] pag. 1, line 2-3. We changed the title.
2- Abstract of the paper is short and must be amended by some numerical values from the key results
[Answer 2] Pag. 1 line 28-33. we have modified the abstract by inserting the most striking data.
3 - Key words: ( cancer; amino acids; colon; human; diet; mice ) is can be amended, please use the best words for indexing purposes. please add the name of the amino acids if possbile and combine (cancer + colon + human) to be (human colon cancer) and add a key word for the (HCT116 cells), please improve the key words
[Answer 3] Pag. 1 line 38. We believe we've improved the keywords. It's not possible to include the names of all the AAs in the blend, so we've limited ourselves to "essential amino acids.".
4 - The intro is long and must be shortened to be more concrete and explore the novelty and research gap for this study
[Answer 4] Pag 2-3. We have shortened the introduction, intervening in several points, to better explain the novelty and potential usefulness of our experimental approach.
5 - The last paragraph in the intro representing the aim of the study needs to be rewritten as it looks like a conclusion, it should be a clear aim describing the goal of the study and how you planned to acheive it
[Answer 5] Pag. 3. We removed the last paragraph as it contained the conclusions.
6 - Some long parts in the into comes with a single refernces. Please add and cite appropriate refernces
[Answer 6] Pag 2-3. We have modified the introduction in several places, shortening the paragraphs and their respective bibliographic references accordingly.
7 - Images in figure 8 are captured at different microscopic powers, please provide equal powers for appropriate comprisons
[Answer 7] Pag. 8, line 227. We replaced photos 8A and 8E, making the magnification the same as the other photos.
8 - what was the weight of mice at the begin of the experiment
[Answer 8] In the subcutaneous injection group, the initial weight was 22.66 ±1.51 (StD) and 23.07 ±1.08 (EAARD) as indicated in Figure 4A (pag. 5, line 151). In the intraperitoneal injection group, the initial weight was 23.33 ±1.01 (StD) and 23.08 ±0.98 (EAARD) as indicated in Figure 9A (pag. 9, line 249).
9 - a diagram showing the step wise actions and prpearation for the behavior task is a worth addition to this paper
[Answer 9] I apologize, but I don't understand what's required. We've summarized the protocol's main steps and results in the graphical abstract.
10 - In methods , Mention in details the housing conditions
[Answer 10] Pag. 14, line 436-437. Housing conditions are described in Materials and Methods, under Animals, Lines, etc. We have specified some characteristics of the housing room. If you deem other parameters necessary, please let us know.
11 - how authors were keen to reduce animal suffering
[Answer 11] Pag. 14 In vivo experiments, the standards set by national and international laws regarding animal protection and welfare for scientific purposes were applied (see line 415-418). In particular, after inoculation of tumor cells, the animals were assessed daily by the responsible veterinarian to determine behavioral/postural changes indicative of suffering, as stated in the Materials and Methods, Animals section (line 443-444).
12 - Mention the methods of anesthesia and the type of anesthetic in details
[Answer 12] Pag. 14-15, line 440-446. Subcutaneous and intraperitoneal injections do not require anesthesia. The animals were sacrificed by cervical dislocation, which does not require anesthesia.
13 - Authors should mention whether any data sets did not follow normal dist after Shapiro Wilk test
[Answer 13] Pag. 15, line 476-477. Since the Shapiro-Wilk test showed a significant departure from normality, we used the Welch t-test, which does not require the assumption of equal variance across populations.
14 - Use appropriate abbreviations for minutes, seconds...etc
[Answer 14] We have checked the text for correct abbreviations of the units of measurement used.
15 - Authors should give the source of chemicals, kits and antibodies completely and consistently (code, company, town, state and country) & version for software
[Answer 15] Pag. 14, line 409-411. The required information regarding the main chemicals used and the antibody codes are described in the Materials and Methods sections of the text. We have added the missing ones.
16 - Every abbreviation in figures should be explained in the figure legend to be self explanatory & stands alone.
[Answer 16] We have explained the abbreviations in the figure captions. Pag. 4 line 115,116,126; Pag. 5 line 138,139; Pag. 6 line 154,163; Pag 7, line 188; Pag. 8 line 228,229; Pag. 9 line 250-254; Pag. 10 line 272,273.
17 - Mention "n" in each illustration individually
[Answer 17] In the figure legend at Pag. 5 line 153; Pag. 6 line 165; Pag. 7 line 175; Pag. 9 line 251, 253, 255; Pag 10, line 276; we have mentioned "n".
18- please mention why amino acids inhibited tumor growth in abstrat
[Answer 18] Pag 1, line 31-33. We have specified why AAs prevent the proliferation of tumor cells.
19- please discuss thoroughly why amino acids inhibited tumor growth in discussion
[Answer 19] We would be happy to describe in detail the mechanism by which this AA mixture inhibits tumor growth. However, as stated in the text and in a previous comment, the data collected so far are based on morphological evidence and the variation of certain immunohistochemical parameters, which certainly do not allow for a comprehensive and definitive explanation of the mechanism of tumor metabolic inhibition, which is likely very complex at the molecular level. As we have discussed, we can only highlight some parallels and confirmations with previous publications, which are very few that use complete EAA mixtures like ours, and propose hypotheses. The aspect we believe to be important with the available data, albeit on an experimental model but with human cells, consists in proposing a paradigm shift in the nutritional approach to cancer patients, where complete and balanced amino acid supplementation should no longer be seen as a dangerous tumor promoter, but as a tool that improves the patient's quality of life by maintaining muscle mass and the metabolism of healthy cells, instead creating an environment unfavorable to the development of tumor cells.
20 - please add the limitation and future directions of this study at the end of the conclusion
[Answer 20] Pag.16, line 489-502. we moved the limitations after the conclusions.
21- for histology stains: how many sections were taken from each animal and how many animals from each group?
[Answer 21] Pag 15, line 459,460. Sections were obtained from 5 animals per group. Five sections were obtained from each tumor, spaced approximately 100 μm apart.
Round 3
Reviewer 2 Report
Comments and Suggestions for Authors
The revised version of paper titled (Inhibitory effects of special mixture of amino acids on human colon cancer cell (HCT116) in vitro and in vivo.) Wss improved compared to the original submitting
Iam glad to accept this form in IJMS